# Utilizing Previously Grafted Sinus as Intraoral Donor Site for Successful Augmentation in Peri-Implant Osseous Defect: A Case Report

**DOI:** 10.3390/medicina58050598

**Published:** 2022-04-27

**Authors:** Won-Bae Park, Jun-Sang Park, Ji-Young Han, Philip Kang

**Affiliations:** 1Department of Periodontology, School of Dentistry, Kyung Hee University, Private Practice in Periodontics and Implant Dentistry, Seoul 02771, Korea; njkysh@naver.com; 2Division of Periodontics, Section of Oral, Diagnostic and Rehabilitation Sciences, Columbia University College of Dental Medicine, New York, NY 10032, USA; jp3949@cumc.columbia.edu; 3Department of Periodontology, Division of Dentistry, College of Medicine, Hanyang University, Seoul 04763, Korea; hjyperio@hanyang.ac.kr

**Keywords:** autogenous bone graft, deproteinized bovine bone mineral, guided bone regeneration, maxillary sinus floor augmentation

## Abstract

The purpose of this case report is to introduce a novel guided bone regeneration (GBR) technique that utilized bone harvested from previously grafted maxillary sinus with deproteinized bovine bone mineral (DBBM) 16 years ago. The patient is a 63-year-old male with hopeless maxillary right molars due to severe bone loss. Two months after the extraction, two bone blocks were harvested with a trephine drill from the lateral wall. One was used for histologic analysis and the other was crushed into particulate forms, which was used for a GBR procedure around an implant at the time of implant placement. The grafted site was then covered with a resorbable collagen membrane. The histological specimen showed newly-formed bone containing residual DBBM particles. The DBBM in the harvested bone was mostly resorbed; DBBM particles comprised only 3.6% of the total bone volume. The final prosthesis was delivered six months post-operatively. No change in crestal bone around the implant was observed throughout the 2 year follow-up period. Within the limitation of the present case report, previously grafted sinus can be a good donor site for further harvesting for a successful GBR procedure.

## 1. Introduction

Maxillary sinus floor augmentation is a predictive procedure that enables successful implant placement in the pneumatized sinus and severely atrophic maxilla [1]. The need for maxillary sinus augmentation procedures have increased due to the increase in the number of implants placed. In turn, the biomaterials used in sinus augmentation have also diversified and evolved. Along with maxillary sinus augmentation procedures, bone graft substitutes used for bone augmentation procedures include autograft, allograft, xenograft, and synthetic bone graft. Bone graft materials primarily function as scaffolds contributing to maintenance and stability of space for osteogenic cells [2]. Collectively, allografts, xenografts, and alloplasts are known as “bone substitutes.” This classification is based on differences in biological origins and thus places emphasis on differences among the materials with regard to undesirable immunoreactions or transmission of unknown pathogens from the graft material [2]. The traditional classes of bone graft materials include autogenous bone, allografts, xenografts, and alloplasts. Allografts consist of freeze-dried human bone with or without demineralization (DFDBA, demineralized freeze-dried bone allograft; FDBA, freeze-dried bone allograft). Xenografts are usually formed from bovine bone-derived materials. Various types of alloplasts have been developed that utilize hydroxyapatite (HAp) or other calcium phosphate compounds such as beta-tricalcium phosphate (β-TCP) [2]. Whether the donor site is intraoral or extraoral, autograft has long been considered the golden standard among the different types of bone graft substitutes [2]. Autografts have vital cells with osteogenic potential which is either osteoconductive or osteoinductive [3]. Currently, the use of xenograft and synthetic bone have increased and their clinical and histological results are reported to be similar to autograft [1,4]. The use of autograft is limited due to the need for additional surgery, donor site morbidity and the size of harvested bone required. If autografts can overcome these limitations, it will maintain the position of the golden standard among bone graft substitutes [5].

Additional surgeries can be avoided if the donor bone is obtained from a site adjacent to the surgical site. When placing implants in the posterior maxillary region and additional bone augmentation is required, the use of a bone graft from a previously grafted ipsilateral maxillary does not require additional surgery for autogenous bone collection. To the knowledge of the authors, this is the first documented case report to explore the use of a bone graft obtained from a previously grafted maxillary sinus for guided bone regeneration (GBR) procedure around an implant at the time of implant placement.

The purpose of this case report is to show that a previously grafted maxillary sinus containing deproteinized bovine bone mineral (DBBM) can be used as an intraoral autograft donor site when performing GBR procedure around an implant at the time of implant placement.

## 2. Case Description

The patient is a 63-year-old non-smoking male suffering from severe chronic periodontitis. The patient underwent a bilateral maxillary sinus augmentation using DBBM (Bio-Oss^®^, Geistlich, Biomaterials, Wolhuson, Switzerland) 16 years ago. Implants were placed in the posterior maxillary region at the time of the sinus augmentation as shown in the pre-operative panoramic radiograph in Figure 1a. The lateral window site was covered with Gore-tex membrane (W.L Gore & Associates, Flagstaff, AZ, USA). 6 months after the surgery, the implants were uncovered and the Gore-tex membrane was removed. Two months after the second stage procedure, the final prosthesis was delivered. The patient was initially compliant for maintenance twice a year but eventually discontinued treatment at the clinic. After 16 years, the patient returned to the clinic due to hypermobility of the periodontally-involved maxillary right 1st molar. The maxillary right 1st molar was extracted and planned to be restored with an implant. 2 months post extraction, sufficient amount of bone was present in the previously augmented maxillary sinus as depicted in Figure 1b. However, ridge deficiency was observed in the edentulous ridge. In Figure 1c, the maxillary left 1st premolar implant and three mandibular right molar implants were also removed due to peri-implantitis.

### 2.1. Surgical Procedures

Prophlyaxis 2 g of amoxicillin was administered 1 h before surgery. Under local anesthesia with 2% lidocaine containing 1:100,000 epinephrine, the buccal and palatal mucoperiosteal flap was reflected for implant placement as shown in Figure 2a. Vertical bony defects were present in the partially healed extraction socket. Autogenous bone was harvested twice using an Ø 4.0 mm trephine drill (Zimmer Biomet, Warsaw, IN, USA) from the lateral surface of the maxilla apical to the implants as illustrated in Figure 2b. One of the two bone cores was fixed in neutral buffered formalin solution (Sigma-Aldrich, St. Louis, USA) for biopsy and the other was crushed into a size of 1–3 mm to be used as bone graft shown in Figure 2b,c respectively. A SLA-textured implant (Ø 4.3 × 12 mm Implantium, Suwon, Korea) was placed in the extraction socket. Due to the ridge deficiencies, the implant platform was placed supracrestally to accomadate for ideal implant placement as depicted in Figure 2e. The crushed harvested bone was placed into the mesial bony defect around the implant and supracrestally around the implant up to the implant platform as seen in Figure 2f. The site was covered with an resorbable collagen membrane (Genoss, Suwon, Korea) as shown in Figure 2g respectively. The flaps were sutured tension-free using 4-0 nylon, but as shown in Figure 2h, the membrane was partially exposed at the extraction socket site; primary closure was not achieved.

Postoperative antibiotics (Cefaclor 375 mg, Yuhan Pharmaceutical Co., Seoul, Korea) and a non-steroidal anti-inflammatory drug (Etodol^®^ 200 mg, Yuhan Pharmaceutical Co., Seoul, Korea) were prescribed for 10 days. The patient was advised to rinse with 0.12% chlorhexidine solution (Hexamedine, Bukwang Pharmaceutical, Seoul, Korea) twice a day for 2 weeks. Sutures were removed after 7 days. Healing was uneventful.

### 2.2. Micro-CT and Histologic Evaluation

A bone core was fixated in a neutral buffered formalin solution (Sigma Aedrich, St. Louis, MO, USA) for 2 weeks. The specimen was dehydrated in an ethanol solution. Specimen was subjected to micro-CT (Skyscan 1173, Kontich, Belgium) analysis. The scanner had a tube voltage of 130 KV and a resolution of 14.91 μm (intensity 60 μA). The specimen was sectioned for histopathological examination followed by a hematoxylin and eosin stain (H-E stain). Histological analysis was performed using a light microscope (BX-51, Olympus Optical, Tokyo, Japan). The microcomputed tomography (micro-CT; SkyScan 1173, Bruker, Kontich, Belgium) test showed that DBBM particles were interspersed between well-organized trabecular patterns and corticalization of the bony plate was observed in the buccal region in Figure 3a. In Figure 3b, DBBM was scattered in the bone core. Quantitative analysis by micro-CT was performed on the bone core. The total bone volume of the bone core measured in mico-CT was 38.83 mm^3^ of which new bone formation was 37.48 mm^3^ (96.4%) and the DBBM was 1.35 mm^3^ (3.6%). In the histologic analysis, highly vascularized soft tissue was seen between trabecular bones as illustrated in Figure 3c. DBBM particles that were not absorbed was surrounded by mature bone tissue was observed as shown in Figure 3d. The presence of osteoclasts and immature bone were also observed in Figure 3e. The bone core consisted of a small amount of DBBM with the majority consisting of regenerated bone containing vital osteogenic cells.

### 2.3. Postoperative Evaluation

In Figure 4a, complete radiographic bone fill was observed in the immediate post-operative panoramic radiograph. The implant was uncovered at 6 months post-operatively. After reflecting the buccal mucoperiosteal flap, the implant was surrounded by hard tissue up to the implant platform as illustrated in Figure 4b. The final prosthesis was delivered 2 months after the healing abutment was inserted. Two years after the delivery of the final prosthesis, a panoramic radiograph, Figure 4c, was taken to confirm the radiographic bone fill.

## 3. Discussion

This case report showed that a previously augmented maxillary sinus can be a good donor site for GBR procedure around an implant at the time of implant placement. Histologic analysis showed that the bone core harvested from the maxillary sinus floor augmentation site, which was performed 16 years ago using DBBM, was mostly newly-formed bone with only 3.6% of DBBM remaining. This indicates that most of the DBBM was resorbed over time and replaced with newly-formed bone.

The maxillary sinus floor augmentation was introduced over 40 years ago and has allowed for successful implant placement in posterior maxilla regions with either limited residual bone height or with a pneumatized sinus. The survival rate of implants placed in the augmented maxillary sinus sites is more than 90% and its utilization is gradually increasing [6]. Although autograft is considered the golden standard among bone graft substitutes [2], other bone graft substitutes, such as allograft, xenograft, and synthetic bone, also show good clinical, histological, and radiolographic results in maxillary sinus floor augmentation [1,4]. Non-autogenous bone grafts have the advantage of reducing morbidity that may occur during bone harvesting. Unfortunately, these bone graft substitutes do not have vital osteogenic cells like autogenous bone grafts.

One of the shortcomings of intraoral autograft is the limited amount of bone that can be harvested [5]. Intra-oral sites used are symphysis, ascending ramus, zygomatic alveolar crest, maxillary tuberosity, and tori [7]. Bone from previously augmented sinuses can be harvested in large amounts as a core type or a block bone type. Furthermore, there is no risk of damage to blood vessels and nerves during harvesting. In particular, potential damage to the posterior superior alveolar artery, which could severely bleed during lateral window preparation, is not a risk factor during graft harvesting. In addition, if an implant is placed in a previously grafted sinus, it is possible to collect the core bone at a site in-between the implants.

Strictly speaking, intraoral autograft harvested from a previously grafted sinus is a composite graft in which DBBM is dispersed throughout rather than pure vital bone. In this case, only DBBM was used to augment the maxillary sinus floor. DBBM was properly condensed at the sinus floor to prevent the formation of dead space. However, during the healing process, new bone was formed between DBBM particles. Throughout the 16 years, the maxillary sinus bone graft was remodeled with continuous cell turnover. Micro-CT of the bone core obtained from the augmented sinus site revealed that the newly formed bone between DBBM particles occupied more volume than the DBBM particles. Histological findings showed that the trabecular bone and cortical layer were properly arranged, and the interconnection between the bone graft particle and the bone was appropriate. DBBM was scattered in the core bone and arranged around mature and immature bone. Osteoclasts were found around DBBM suggesting that DBBM resorption was still ongoing.

Another important concern with autogenous bone grafts is volumetric change [8]. The resorption rate of intramembraneous bone such as the maxilla is lower than that of embryologic bone such as the iliac crest bone [9]. On the other hand, DBBM’s slow resorption over time is considered to be an advantage of DBBM but there are very few human histologic reports on DBBM resorption over time [10,11,12,13]. Guarnieri et al. reported that there was 33.58% residual DBBM after 24 months of healing and that there was considerable absorption of DBBM. Iezzi et al. [14] reported that after the anorganic bone matrix was slowly absorbed, leaving only a small quantity of 22.1% after 14 years [14]. However, Mordenfeld et al. said that there was no difference in the volume of DBBM in terms of histomorphometrical analysis 6 months and 11 years after a maxillary sinus floor augmentation procedure [15]. As such, there have been several disagreements regarding the rate at which DBBM is absorbed. In the present case, a significant portion of DBBM was resorbed and replaced with the newly-formed bone after 16 years; only 3.6% of DBBM remained. In comparison to previously reported studies, the presented case had a significantly smaller amount of residual DBBM.

The findings from this case report showed that the intraoral autograft harvested from a previously augmented sinus was a reasonable bone graft substitute. There is a limitation to some of the conclusions in this case report, in particular the number of the cases. Better clinical guidance can be provided if more cases are documented and reported.

## 4. Conclusions

Within the limitation of this present report, a previously augmented sinus site is a reasonable intraoral graft donor site for a successful GBR procedure.

## Figures and Tables

**Figure 1 medicina-58-00598-f001:**
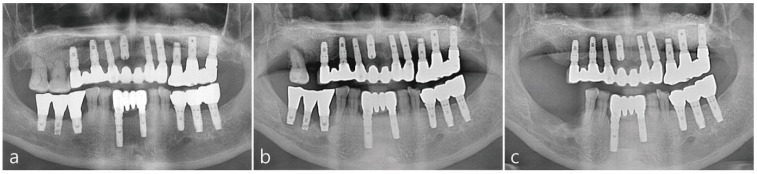
(**a**) Pre-operative panoramic radiograph (**b**) Panoramic radiograph taken after the extraction of the maxillary right first molar (**c**) Panoramic radiograph taken after the extraction of the maxillary right second molar and removal of right posterior implants.

**Figure 2 medicina-58-00598-f002:**
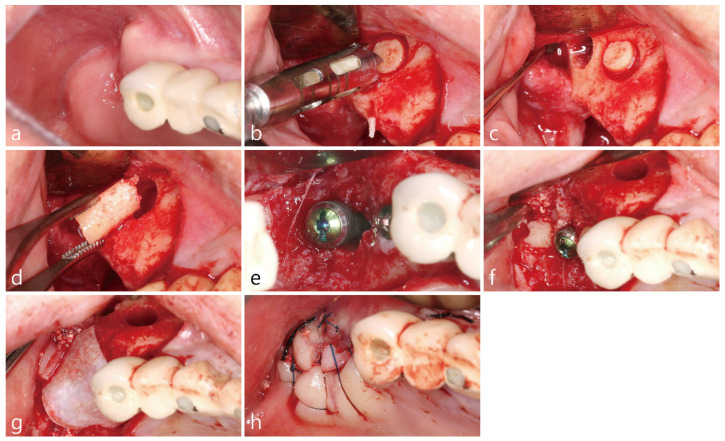
(**a**) Intra-oral photo of upper right maxillary quadrant 2 months post extraction (**b**) Utiliziation of a trephine bur to collect bone from previously augmented right maxillary sinus. (**c**) Bone was harvested twice for histologic analysis and utilization for new implant site (**d**) Intra-oral photo of the size of bone core obtained. (**e**) New implant was placed in previous extraction site and the implant platform was supracrestal to the ridge to allow for vertical bone augmentation. Vertical intra-osseous defect was present on the mesial surface of the implant. (**f**) Harvested bone core was crushed and placed into the intra-osseous defect. (**g**) The surgical site was covered with an resorbable collagen membrane. (**h**) The flap was closed tension-free. The membrane was partially exposed at the extraction site.

**Figure 3 medicina-58-00598-f003:**
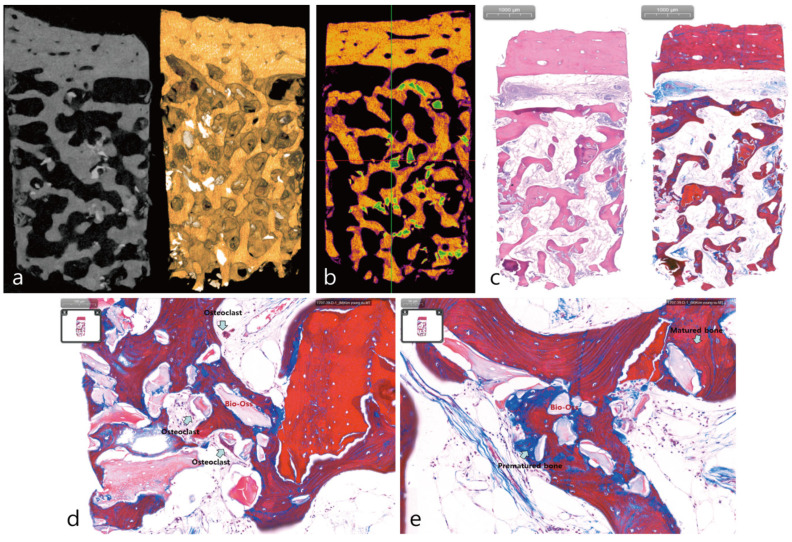
Micro-CT images and histologic finding of the bone core. (**a**) Micro-CT findings. Deproteinized bovine bone mineral(DBBM)particles were interspersed between well-organized trabecular patterns. In the buccal side, a bony plate with good corticalization was observed. (**b**) The appearance of DBBM scattered in the bone core. (**c**) Histologic finding. There was a presence of highly vascularized soft tissue between the trabecular bones (left: H-E stain, right: MT stain). (**d**) DBBM particles and mature bone tissue (red color) were observed. Osteoclasts were also observed around DBBM particles (arrow). (**e**) A newly formed premature bone (blue color) and matured bone (red color) was observed around the DBBM particles (arrow).

**Figure 4 medicina-58-00598-f004:**
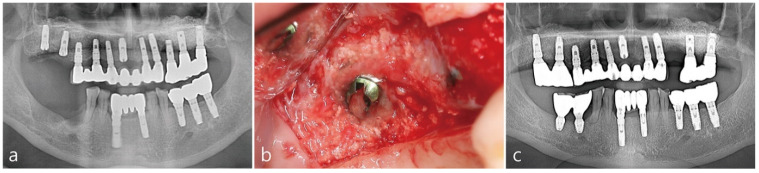
(**a**) Immediate post-operative panoramic radiograph. Intraoral bone augmented around the implant was observed. (**b**) Intra-oral photo taken 6 months post-operatively during the second stage procedure. (**c**) Panoramic radiographs taken two years after prosthesis delivery showed radiographic bone fill around the implant was maintained.

## Data Availability

Not applicable.

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
