# Peer review of "Utilizing Previously Grafted Sinus as Intraoral Donor Site for Successful Augmentation in Peri-Implant Osseous Defect: A Case Report"

_medicina, 2022, doi:10.3390/medicina58050598_

Round 1

Reviewer 1 Report

The case report entitled “Utilizing Previously Grafted Sinus as Intraoral Donor Site for Successful Augmentation in Peri-implant Osseous Defect: A case Report” aimed introduce a novel guided bone regeneration (GBR) technique that utilized bone harvested from previously grafted maxillary sinus with deproteinized bovine bone mineral (DBBM) 16 years ago.

The paper is in line with journal’s aim, moreover, Authors have well revised several issues; however, I ask authors to add some key concepts.

  • In the introduction section, the authors should add previous findings on selected antimicrobial agents also on other body districts (e.g., CHX is an agent able to inhibit plaque formation and remains the safest and most effective antimicrobial agent used for the reduction of microorganisms in the oral cavity, please see and discuss PMID: 31204452 and PMID: 30410934)
  • The limits of the study should be included in the paper
  • Conclusions cannot be reduced to a sentence: you must improve them highlighting the limits and the future insights pointed out from this article.
  • The formatting of the references is not correct, please check the journal instructions for authors
  • Several moderate typos are present in the text, please, amend

Author Response

In the introduction section, the authors should add previous findings on selected antimicrobial agents also on other body districts (e.g., CHX is an agent able to inhibit plaque formation and remains the safest and most effective antimicrobial agent used for the reduction of microorganisms in the oral cavity, please see and discuss PMID: 31204452 and PMID: 30410934)

  • The use of CHX following the surgery was included under Case Description, and the rationale was also added.

The limits of the study should be included in the paper

Conclusions cannot be reduced to a sentence: you must improve them highlighting the limits and the future insights pointed out from this article.

  • The limitation of the presented case reports and suggested recommendations for the future are included in the Discussion section and further modified for improvement.   

The formatting of the references is not correct, please check the journal instructions for authors

  • The reference format has been corrected.

Several moderate typos are present in the text, please, amend

  • Several typos and grammatical errors were identified and corrected.

Reviewer 2 Report

Manuscript ID: medicina-1662229

Title: Utilizing Previously Grafted Sinus as Intraoral Donor Site for Successful Augmentation in Peri-implant Osseous Defect: A Case Report

1.What is the main question addressed by the research?

To report a novel guided bone regeneration (GBR) technique that utilized bone harvested from previously grafted maxillary sinus with deproteinized bovine bone mineral (DBBM) 16 years ago.

2.Is it relevant and interesting?

The article is relevant and interesting.

3.How original is the topic?

The topic is current.

4.What does it add to the subject area compared with other published material?

The authors reported a really interesting case.

5.Is the paper well written?

Yes, the article is well written.

6.Is the text clear and easy to read?

Yes, but minor English editing is required.

7.Are the conclusions consistent with the evidence and arguments presented?

Yes, the conclusions (reported at the end of discussion section) consistent with the evidence and arguments presented but further studies are needed to confirm Authors’ hypothesis.

8.Do they address the main question posed?

Yes, the Authors addressed the main question posed.

Other comments:

Author should follow SCARE guidelines [Agha, Riaz A., et al. "The SCARE 2018 statement: updating consensus Surgical CAse REport (SCARE) guidelines." International Journal of Surgery 60 (2018): 132-136].

  • English language: Minor spell check required
  • Summary of abbreviations required.
  • Introduction: This section needs few improvements. For example, Authors may include a brief sentence on osseointegration and factors that can affect it based on the following reference: <<Bone quality and quantity could influence fixture osseointegration and the success of implant-prosthetic procedures [PMID: 32475099]>>. The Authors may improve this section on the theme of alternative instruments for crestal sinus augmentation, implant site preparation, and placement. Allow me to suggest a relevant references to include: “PMID: 35055423”.
  • Case description: This section has been properly prepared.
  • Discussion: This section has been properly prepared.
  • Conclusion: Please state a conclusion or remove this section.

After making the indicated changes, I am available for a second round of peer review.

Thanks for the opportunity to review this manuscript.

Author Response

Dear Reviewer,

We appreciate the time and effort you have dedicated to refining our manuscript.  We have edited the manuscript addressing your suggestions and point-by-point responses are written below.  Thank you.

Author should follow SCARE guidelines [Agha, Riaz A., et al. "The SCARE 2018 statement: updating consensus Surgical CAse REport (SCARE) guidelines." International Journal of Surgery 60 (2018): 132-136].

  • The manuscript has been reformatted to follow the SCARE guidelines.

English language: Minor spell check required

  • Spelling and grammatical errors have been corrected.

Summary of abbreviations required.

  • All abbreviations have been summarized.

Introduction: This section needs few improvements. For example, Authors may include a brief sentence on osseointegration and factors that can affect it based on the following reference: <<Bone quality and quantity could influence fixture osseointegration and the success of implant-prosthetic procedures [PMID: 32475099]>>. The Authors may improve this section on the theme of alternative instruments for crestal sinus augmentation, implant site preparation, and placement. Allow me to suggest a relevant references to include: “PMID: 35055423”.

  • Thank you for the suggestions.  We have included a few sentences on factors influencing osseointegration, the long-term stability of implant-supported prostheses, and proposed alternatives.

Case description: This section has been properly prepared.

Discussion: This section has been properly prepared.

Conclusion: Please state a conclusion or remove this section.

Reviewer 3 Report

Dear Authors

I read that this case report looks good and has good pieces of evidence for the DDM from a bovine source. I have a few suggestions to improve this paper with some recent information. Please follow the below comments. 

1) Abstract is good but keywords are not selected on the basis of the work reported. Please improve it

2) Introduction is not well written. Authors need to improve it carefully. During the improvement of the introduction, paragraph authors can use the information from this published paper and strengthen the introductory paragraph. ( https://www.mdpi.com/1420-3049/26/10/3007 )

3) I am wondering why the authors do not discuss " The CARE guidelines (for CAse REports)" for their work? Need an explanation for it and if possible try to upload this guideline checklist as a supplementary file. 

4)  In discussion authors have to add few more clinically oriented studies outcomes and link with their outcomes. 

5) Conclusion : authors not provide this part. Add conclusive remarks as well as future suggestions. 

Author Response

Dear Reviewer,

We appreciate the time and effort you have dedicated to refining our manuscript. We have edited the manuscript addressing your suggestions and point-by-point responses are written below. Thank you.

1) Abstract is good but keywords are not selected on the basis of the work reported. Please improve it

- More keywords that are relevant to the submitted case report have been added

2) Introduction is not well written. Authors need to improve it carefully. During the improvement of the introduction, paragraph authors can use the information from this published paper and strengthen the introductory paragraph. ( https://www.mdpi.com/1420-3049/26/10/3007 )

- Introduction has been modified based on your suggestions.

3) I am wondering why the authors do not discuss " The CARE guidelines (for CAse REports)" for their work? Need an explanation for it and if possible try to upload this guideline checklist as a supplementary file. 

- The manuscript has been reformatted to follow the SCARE guidelines for case reports.

4)  In discussion authors have to add few more clinically oriented studies outcomes and link with their outcomes. 

5) Conclusion : authors not provide this part. Add conclusive remarks as well as future suggestions. 

In the Conclusion section, additional comments on the limitation of the presented case report and future direction have been included.  Due to the limited availability of published data on the presented surgical technique, unfortunately, no additional studies could have been presented.

Thank you very much. 

Round 2

Reviewer 2 Report

After the changes made the article may be suitable for publication.

Author Response

Thank you very much for your valuable insights.  

Reviewer 3 Report

Dear Authors

well revised but I have noticed my second comment is ignored. 

2) Introduction is not well written. Authors need to improve it carefully. During the improvement of the introduction, paragraph authors can use the information from this published paper and strengthen the introductory paragraph. ( https://www.mdpi.com/1420-3049/26/10/3007 )

  • Introduction has been modified based on your suggestions.

This comment is good for your paper improvement. 

Author Response

Dear Reviewer,

Thank you very much for your guidance.  As you suggested, more information on various types of bone graft substitutes and descriptions have been added in the introduction.